| 1<br>2 | Geomorphic Threshold Estimation for Gully Erosion in the Lateritic Soil of Birbhum, West Bengal, India                         |
|--------|--------------------------------------------------------------------------------------------------------------------------------|
| 3      | Sandipan Ghosh <sup>1</sup> , Sanat Kumar Guchhait <sup>2</sup>                                                                |
| 4<br>5 | <sup>1</sup> Assistant Professor, Dept. of Geography, Chandrapur College, Chandrapur – 713 145, Barddhaman, West Bengal, India |
| 6      | <sup>2</sup> Professor, Dept. of Geography, The University of Burdwan, Barddhaman – 713 104, West Bengal, India                |
| 7      | Correspondence to: Sandipan Ghosh (sandipanghosh19@gmail.com)                                                                  |
| 8      |                                                                                                                                |
| 9      |                                                                                                                                |
| 10     |                                                                                                                                |
| 11     |                                                                                                                                |
| 12     |                                                                                                                                |
| 13     |                                                                                                                                |
| 14     |                                                                                                                                |
| 15     |                                                                                                                                |
| 16     |                                                                                                                                |
| 17     |                                                                                                                                |
| 18     |                                                                                                                                |
| 19     |                                                                                                                                |
| 20     |                                                                                                                                |
| 21     |                                                                                                                                |
| 22     |                                                                                                                                |
| 23     |                                                                                                                                |
| 24     |                                                                                                                                |
| 25     |                                                                                                                                |
| 26     |                                                                                                                                |
| 27     |                                                                                                                                |

# 1 Abstract

2 It is a growing concern that accelerated soil erosion which aggravated by the processes of water erosion (rainsplash, inter-rill, rill, gully erosion and soil piping) in the tropical region and soil loss through 3 crop harvesting represent a seriously threat to soil resource and the ecosystem services that it prevails. The 4 5 present study examines a quantitative approach to predict critical conditions and locations where gully heads might develop in the lateritic terrain, located at eastern plateau fringe of Rajmahal Basalt Traps, India 6 7 (Birbhum, West Bengal). The modern concept of geomorphic threshold is applied here on the issue of gully erosion hazard (specially in Indian context) to identify the critical slope of gully head (S) and upstream 8 drainage area (A) with a core relationship of  $S = a A^{-b}$ . On the basis of 118 gully heads we have derived 9 statistically significant relationships between slope and drainage area (r = -0.55), overland flow (Q) and 10 slope length (L) (r = 0.69), relative shear stress ( $\tau$ ) and slope (r = 0.92), overland flow detachment rate (H) 11 and eroding force of overland flow (F) (r = 0.98). The established S – A critical relationship, as geomorphic 12 threshold line, is expressed as  $S = 17.419 \text{ A}^{-0.2517}$ , above which gully initiation occurs on the laterites. This 13 equation can be used a predictive model to locate the vulnerable un-trenched slopes (i.e. potential gully 14 15 erosion locations) in other lateritic areas of West Bengal. The constant b value (0.2517) and Montgomery -Dietrich Envelope suggest a relative dominance of overland flow (52.51 % of gully heads) in the processes 16 of gully erosion. The result of revised MMF model reflects soil loss of 2.33 to 19.9 kg m<sup>-2</sup> yr<sup>-1</sup> due to 17 overland flow erosion. 18

# 19 Keywords: Gully, laterite, geomorphic threshold, overland flow, M – D Envelope

  - **)**4

1

2

# 1. Introduction

3 Soil is a fundamental resource that provides a number of ecosystem services and it is the dynamic medium on which we produce 99 per cent of our food in addition to fodder, fibre, raw materials and biofuels 4 (Bilotta et al., 2012; Brevik et al., 2015; Smith et al., 2015). Soils contribute to basic human needs like food, 5 clean water and clean air, and are a major carrier for biodiversity (Keesstra et al., 2016). Soils have critical 6 7 relevance to current global issues such as food and water security, climate regulation, land degradation and desertification (Zdruli et al., 2010; Montanarella et al., 2016). Nowadays intensive soil erosion and land 8 degradation have raised question on the environmental sustainability and the actions of land users. The 9 global population is predicted to reach 9 billion by 2050; in combination with changes in dietary behaviour, 10 11 a large net increase in productivity and agricultural area is needed (Brevik et al., 2015). But continuous loss 12 of land and soil cover will make the above situation more critical, due to immense pressure on the soil 13 resource. If we continuously loss soil, then we shall immediately face hurdle to achieve food security in the 14 developing countries like India where agriculture till now is the socio-economic base. It is learned that soil resource is being lost from the land areas 10 to 40 times faster than the rate of soil renewal imperiling future 15 16 human food security and environmental quality (Pimentel, 2006; Pimentel and Burgess, 2013). In South 17 Asia, annual loss in soil productivity was estimated at 36 million tons of cereal equivalent valued at US \$ 18 5,400 million by water erosion, whereas among the various estimates of soil erosion costs between 1933 and 2010, the highest figure was US \$ 95.5 billion a year for the European Union and US \$ 44 billion a year for 19 the United States (Eswaran et al., 2001; Telles et al., 2011). Therefore, a fundamental greatest priority is that 20 we should quantitatively assess the soil erosion, then control the further degradation of soils and restore the 21 22 soil productivity that are already degraded in the erosion-prone regions where people are most vulnerable.

23 Soil erosion is continuously triggering the land degradation and expansion of wastelands in many areas of world (Cerda et al., 2009; Zhao et al., 2013; Gabarron-Galeote et al., 2013; Mandal et al., 2013; 24 Lieskovsky and Kerderessy, 2014; Mekonnen et al., 2014a, 2014b, 2015; Dai et al., 2015; Erkossa et 25 al.,2015; Prosdocimi et al., 2016; Novara et al. 2016). Soil erosion is a severe geomorphic hazard 26 traditionally associated with livelihood in the tropical and semi-arid areas, influencing long-term effects on 27 28 soil productivity and sustainable agriculture (Morgan, 2005). Globally soil erosion leads to environmental problems through sedimentation, pollution, increasing flooding and also growing 75 to 80 percent of carbon 29 content into the atmosphere (Lal, 1992, 1995; Blanco-Canqui and Lal, 2012). 30

Land degradation through soil erosion is now recognized as an important environmental issue in 31 India (Sinha and Joshi, 2012; Pani & Carling, 2013) and gully erosion is the extreme form of accelerated soil 32 erosion, affecting about 3.975 million ha of land in India (Yadavand Bhusan, 2002; Pathak et al., 2006; 33 Singh et al., 2015). In India, 165,912 km<sup>2</sup> of land (5.58 percent of total land) is vulnerable to very high 34 desertification and land degradation (Eswaran et al., 2001).Singh et al. (1992) estimated that soil erosion 35 took place at a rate of exceeding 40 t ha<sup>-1</sup> yr<sup>-1</sup> in the ravines and badlands of India. It is estimated that soil 36 erosion is taking place at average rate of 16.35 ton ha<sup>-1</sup> yr<sup>-1</sup>in India and about 29 percent of total eroded soil 37 is lost permanently to the sea and 10 percent of it is deposited in reservoirs (Narayana et al., 1983). Severe 38 39 soil loss of Damodar River catchment (eastern part of India) increases sediment yield from 1729 to 2387 m<sup>3</sup> km<sup>-2</sup> yr<sup>-1</sup> and it escalates the siltation of Panchet reservoir at a rate of 0.033 to 0.047 cm per year (Majumder 40 et al., 2012). In the humid subtropical region of India soil erosion (about 15 million tonnes per year) leads to 41 low crop productivity and an annual loss of 13.4 million tonnes in the production of crops due to water 42 43 erosion equivalent to about \$2.51 billion (Bhattacharyya et al., 2007; Sharda et al., 2010 and 2013). The extreme form of soil erosion is gully erosion which represents a major sediment producing process, 44

generating between 10 and 95 percent of total sediment mass at catchment scale whereas gully channels 1 2 often occupy less than 5 percent of total catchment area (Poesen et al., 2003; Valentin et al., 2005; Poesen, 2011). In the lateritic region of West Bengal (about 7,700 km<sup>2</sup>) land degradation is acute because of aberrant 3 weather, drought, ferruginous crusting, rill and gully erosion, NKP deficiency, low water holding capacity 4 5 and wide range of land use conversion (Jha and Kapat, 2009 and 2011; Shit and Maiti, 2012 and 2014; Shit 6 et al., 2014). The major districts of lateritic Rarh plain (Birbhum, Barddhaman, Bankura, Purulia and West 7 Medinipur) have 731000 ha of degraded land intensively affecting by gully erosion. Taking steps to prevent or control gully erosion should require no justification. If appropriate measures for gully prevention and 8 9 restoration are to be taken in the lateritic region, it is important to know the threshold conditions in which 10 gullies develop.

11 It is proposed that a gully is relatively deep (> 0.6 m), recently formed eroding channel (with ephemeral flow) that forms on valley sides and on valley floors where no well-defined channel previously 12 existed and it has steep sides, low width-depth ratio and stepped profile (presence of nickpoints), 13 14 characteristically with a headcut (with plunge pool) at the upslope end, dominated by the processes of 15 surface flow, piping and mass movement (Horton, 1945; Brice, 1966; Gregory and Walling, 1973; Schum et 16 al., 1984; Bull and Kirkby, 1997; Knighton, 1998; Erskine, 2005). A high drainage density of rill and gully 17 system deeply incises and dissects the soft-rock terrain to form erosional landscape, called'badlands' which support smallest amount of sparse vegetation in this sterile terrain (Singh and Agnihotri, 1987). The 18 19 badlands of Himalayan Foreland Basin and Chotanagpur Plateau Fringe are believed to have developed due 20 to neo-tectonic activities, strengthening of south-west monsoon and intensive fluvial erosion in Late 21 Pleistocene - Holocene (Ranga et al., 2015 and 2016). In the Rarh Plain of West Bengal (i.e. typical tropical 22 morpho-genetic region, lying west of Bengal Basin), the badlands of lateritic terrain is popularized as 23 'khoai' in Bengali language (Ghosh and Guchhait, 2015).

24 Gully erosion signifies instability in landform (an example of equifinality in the geomorphology) and it is regarded as threshold phenomenon in the landscape (Schumm and Hadley, 1957; Patton and Schumm, 25 1975; Schumm, 1979; Posen et al., 2003; Poesen, 2011; Joshi et al., 2013). There are wide ranges of threshold 26 27 conditions or values (viz., thresholds of hydraulic, rainfall, topography, lithology and land use – land cover 28 control etc.) which are responsible for the initiation of gullies in different environments (Horton, 1945; Patton and Schumm, 1975; Begin and Schumm, 1984; Ebisemiju, 1989; Montgomery and Dietrich, 1994; 29 30 Vandaele et al., 1996; Vandekerckhove et al., 1998; Moeyersons, 2003; Morgan and Mngomezulu, 2003; 31 Posen et al., 2003; Valentin et al., 2005; Hancock and Evans, 2006, Gutierrez et al. 2009; Samni et al., 2009; 32 Dong et al., 2013; Torri and Poesen, 2014; Araujo and Pejon, 2015). Thresholds can be exceeded when input 33 is relatively constant, that is, the external variables remain relatively constant, yet a progressive change of the system itself renders it unstable and failure occurs (Schumm, 1973 and 1979). A geomorphic threshold is 34 35 one that is inherent in the manner within the geomorphic system by changes in the morphology of the landform itself through time, like phenomenon of gully erosion (Schumm, 1973, 1979 and 2004). It is a 36 37 threshold of landform instability that is exceeded either by intrinsic change (e.g. slope steepness and soil cohesion) of the landform itself, or by a progressive change of an external variable (e.g. climate change and 38 neo-tectonic uplift) (Schumm, 1979; Jain et al., 2012). 39

Globally a large amount of research has been dedicated to enhance our understanding of the factors
and mechanisms affecting gully erosion and we have reviewed selected articles about quantitative aspects of
gully erosion (Schumm and Hadley, 1957; Singh and Agnihotri, 1987; Bocco, 1991; Lal, 1992; Bull, 1997;
Howard, 1997; Poesen et al., 1998 and 2003; Sidorchuk, 1999; Yadav and Bhushan, 2002; Torri and
Borselli, 2003; Valentin et al., 2005; Ghimire et al., 2006; Pathak et al., 2006; Kirkby and Bracken, 2009;

Sharma, 2009; Sinha and Joshi, 2012; Poesen, 2011; Joshi et al., 2013; Singh et al., 2015). In many parts of 1 2 world the erosion dynamics of badlands and role of lithology, micro-relief pattern, vegetation and climatic characteristics on the expansion of badlands are precisely studied (Gracia-Fayos et al., 1995; Cerda and 3 4 Garcia-Fayos, 1997; Sole-Benet et al., 1997; Cerda, 1997; Martinez-Murillo et al., 2013). In West Bengal the geomorphic studies of gully erosion are solely concentrated on the erodible lateriteswhich are the most 5 6 vulnerable sites of soil erosion (Kar and Bandyopadhyay, 1974; Basu, 1992; Das and Bandyopadhyay, 1995; 7 Sen et al., 2004; Sarkar et al., 2007; Jha and Kapat, 2009 and 2011; Ghosh and Maji, 2011; Ghosh and Guchhait, 2012; Ghosh and Bhattacharya, 2012; Lenka et al., 2014; Shit and Maiti, 2014; Shit et al., 2015). 8 9 The importance of these researches is mostly concentrated on the processes, estimation and effects of gully 10 erosion. Based on literature review it is found that no research has been performed to understand critical threshold conditions of gully initiation in Indian context. So the prime objective of this experimental work is 11 12 to investigate the geomorphic threshold conditions of permanent gullies on the laterites of West Bengal. It is hypothesized that the gullies over the laterites develop when the geomorphic thresholds (may be extrinsic or 13 14 intrinsic) are transgressed due to either a decrease in the resistance of the materials (i.e. erodibility) or an 15 increase in the erosivity of the runoff or both. Here we have attempted to apply the slope – drainage area (as 16 intrinsic threshold) model in identifying the threshold conditions of gully initiation to identify the dominant process of erosion and to estimate overland flow erosion (as extrinsic threshold) and annual rate of soil loss 17 in the terrain of laterites. 18

19 **2.** Materials and Methods

### 20 2.1 Description of Study Area

The study area (about 176 km<sup>2</sup>) for this experimental work is situated between the adjoin region of 21 western Rampurhat I Block of Birbhum district, West Bengal and eastern Shikaripara Block of Dumka 22 23 district, Jharkhand (encompassing by 24°08' to 24°14' N and 87°38' to 87°44' E). This area is located at 5 to 6 km west of Rampurhat railway station on the highway of NH 114A (around Rampurhat to Dumka 24 road). This geomorphic unit is recognized as elevated interfluve of laterites in between Brahmani (north) and 25 Dwarka (south) rivers and it is the eastern plateau fringe of Rajmahal Basalt Traps (i.e. oldest volcanism 26 27 than Deccan volcanism). The deep ferruginous profile of tropical weathering is the remarkable morpho-28 stratigraphic feature on the basaltic basement, having laterite capping and veneer of lateritic sediments. The elevation of this unit ranges from 20 to 80 m, having average slope of 2.8° towards south-east. The in-situ 29 primary laterites (Pliocene to Early Pleistocene) and *ex-situ* secondary laterites (Early to Late Pleistocene) 30 are simultaneously found in this eastern fringe of Rajmahal Basalt Traps (Early Cretaceous) (Ghosh and 31 32 Guchhait, 2015). The detrital secondary laterites were developed here in loose ferruginous concretions with gravels and pebbles (occasionally over the surface of primary laterites) under prevailing tropical wet - dry 33 palaeoclimate and these were derived by the slope wash or channels from the high level surface of 34 35 weathered primary laterites (from west) and deposited on the piedmont slope of east. This study site is a representative of Rarh (i.e. land of red soil) Bengal (Biswas, 1987) which is affected intensively by rill and 36 37 gully erosion. This lateritic region has lost its soil cover at an alarming rate of 20 to 40 t ha<sup>-1</sup> yr<sup>-1</sup> (Ghosh and Bhattacharya, 2012). The dry tropical deciduous trees (mainly Sal), Acacia plantation, bushes and grassland 38 are the major land covers over the laterites, but intermittently the land is bare with surface crusting (favours 39 high runoff). This erosion-prone barren land is not excessively used for agricultural land use (except used 40 41 for aerodrome purpose) and no land management measures have been taken to check gully erosion.

42 The climate of this region has been identified as sub-humid and sub-tropical monsoon type, receiving 43 mean annual rainfall of 1437 mm. The peak monsoon and cyclonic rainfall intensity of 21.51 (minimum) to 44 25.51 (maximum) mm hr<sup>-1</sup> is the most powerful climate factor to develop this lateritic badland. The recorded

maximum and minim temperature is 45° C (April – May) and 9° C (December – January) respectively, with 1 seasonal variation of 15° to 19° C. The period between mid-June and October is the active erosion phase due 2 to heavy downpours, removing ferruginous sediments from the gullied catchments (Ghosh and 3 4 Bhattacharya, 2012). In the study area the soil series of Bhatina – Raspur – Jhinjharpur (Sarkar et al., 2007) 5 has been developed through the influence of existing geo-climatic settings. Generally, this thin soil is loamy-6 skeletal and hypothermic in nature developing on the barren lateritic wastelands with sparse bushy 7 vegetation. The dark reddish brown sandy clay loam of 0 to 16 cm (A horizon) is developed over weathered secondary laterites. This soil series has weak fine crumb and granular structure (slightly hard, friable and 8 slightly sticky), 2 to 5 mm size of manganese nodules, > 2 mm size of ferruginous nodules with goethite 9 cortex, 30 to 40 percent gravels, excessive drained surface and pH of 5.4. The loose secondary laterite (16 to 10 34 cm thick) over mottle and kaolinite horizon is much prone to overland flow erosion and tunnel erosion. 11

# 12 2.2 Data Collection

The base map of study area is derived from the SOI (Survey of India) topographical sheet of 13 14 1:25,000 scale (72 P/12/NE and 72 P/16/ NW, 1979-80) using Erdas 9.1 and ArcGIS 9.3 software. The regional elevation information is collected from the USGS (United State Geological Survey, earth explorer) 15 16 ASTER (Advanced Spaceborne Thermal Emission and Reflection Radiometer) data of 2011, having 30 m of 17 spatial resolution. All the maps are geo-referenced in UTM (Universal Transverse Mercator) projection with WGS-84 (World Geodetic Survey, 1984) datum. In the GIS (Geographic Information System) framework 18 we have plotted the existing drainage of study area (derived from toposheet) on the ASTER elevation map to 19 depict the regional dissection of water divides. The locations of laterite exposures are mapped on the basis of 20 21 field expeditions, toposheets, survey points of Garmin GPSmap 76CSx receiver(with horizontal accuracy of 22 +- 3m), Google Map and district resource maps of Birbhum and Dumka (Geological Survey of India, 2001). 23 We have employed different empirical equations to quantify the gully erosion and to relate dominant 24 variables. The relationships between variables are examined by performing Pearson's product moment correlation (r), coefficient of determination ( $\mathbb{R}^2$ ), non-linear regression analysis, important statistical tests 25 (viz., Student's t test of correlation coefficient and significance test of standard error of b) and finally 26 27 depicting graphically on the scatter diagrams to get overall picture of erosion system.

The spatial scale to study erosion processes is here plot-scale (10 to 100  $m^2$ ) and field scale (100 to 28 10,000 m<sup>2</sup>). In terms of identifying the geomorphic thresholds in gully initiation, the present experimental 29 work includes the 118 gully heads (both valley-floor and valley-side gullies). To depict the role of ground 30 slope and to identify critical slopes (i.e. potential for gully incision) we have selected 146 valley-side slopes 31 randomly in this lateritic terrain, including gullied and un-gullied slope segments. Sprinter 150 m of Leica 32 Geosystem was used to measure the angle of slope facets. Alongside in few cases (due to obstacles) from 33 ASTER DEM the slope length and angle (usually from gully headcut to water divide) is measured to judge 34 35 the length of surface flow (responsible for gully erosion).Drainage area is calculated from the flow direction 36 and flow accumulation algorithm of ArcGIS 9.3 using drainage lines (digitized from toposheets) and DEM.

#### 37 2.3 Quantitative Techniques and models

A key component of gully network growth and landscape evolution theories(as well as quantitative models for topographically controlled catchment runoff)is prediction of where channels or gullies begin (Montgomery and Dietrich, 1988). Channel initiation by surface processes has been viewed as a threshold phenomenon related to size of contributing area (A) and its slope (S) (Schumm and Hadley, 1957; Patton and Schumm, 1975; Begin and Schumm, 1984; Morgan and Mngomezulu, 2003; Dong et al., 2013; Vandaele et al., 1996; Samni et al., 2009; Araujo and Pejon, 2015; Montgomery and Dietrich, 2004;

Ebisemiju, 1989; Moeyersons, 2003). The relation between critical valley slope and drainage basin area (S =1 a A<sup>-b</sup>, where a = coefficient and b = exponent of relative area) is used as a predictive model to locate those 2 areas of instability within alluvial valleys where gullies will form. The product of S and A ( $SA^2$ ) can be 3 interpreted as a surrogate for stream power and it is used as tool for gully initation point (Montgomery and 4 5 Dietrich, 2004). The idea of taking critical slope as threshold reveals that gully incision demands a minimum 6 runoff discharge in function of slope (Moeyersons, 2003). This erosion system is assumed to be non-linear 7 because the outputs are not proportional to the inputs across the entire range of the inputs (Philips, 2003, 2006 and 2009). A threshold line is drawn through the lower limit of scatter of the points and this line 8 represents, for a given area, a critical value for valley slope above which entrenchment of the laterite should 9 occur. This relationship can be written as  $SA^{b} > T$  (where T = threshold value, i.e. area<sup>b</sup>), defining the limit of 10 threshold value to start gully initiation (Morgan and Mngomezulu, 2003; Torri and Poesen, 2014). An 11 empirically derived equation is developed based on the S – A relationship (S = a  $A^{-b}$ ), relating to the ratio of 12 shear stress applied by the flow and average shear stress of gully channel (Begin and Schumm, 1984). 13

Relative shear stress 
$$(\tau) = A^b S / a$$
 (Eq. 1)

A theoretical division of the landscape into process regimes in terms of log S (X axis) and log A (Y axis) signifies different geomorphic thresholds to gully erosion and the resultant critical threshold line is demarcated as Montgomery – Dietrich (M – D) envelope, through A – S threshold (Montgomery and Dietrich, 1988, 1992 and 1994; Vandekerckhove et al., 2000; Moeyersons, 2003; Samni et al., 2009). To denote the critical tractive or eroding force required for overland flow for initiating a channel, the Du Boy's equation is applied here (Horton, 1945; Kar and Bandyopadhyay, 1974; Ebisemiju, 1989)

21 
$$F = w d \sin \theta$$
 (Eq. 2)

where, F = tractive or eroding force exerted on the slope by overland flow (gm cm<sup>-2</sup>), w = specific weight of water, gm cm<sup>-3</sup> (assumed constant), d = depth of flow in cm and  $\theta$  = gradient of ground slope.

We have applied the Revised Morgan Morgan Finney model (RMMF) to estimate annual overland flow, annual detachment rate of overland flow and annual transport capacity of overland flow. RMMF model separates total erosion process into a water phase and a sediment phase (Morgan et al., 1984; Morgan, 2001 and 2005; Morgan and Duzant, 2008). The analysis is based on annual mean rainfall of 1437 mm, recorded in Rampurhat weather station (controlled by Irrigation and Waterways Department, Government of West Bengal). The estimation is based on the slope length scale study which incorporates the maximum angle of slope with maximum length of slope in the sample catchments of gully heads.

#### 31 **3. Results and Discussion**

#### 32 **3.1 Threshold of Gully Development**

33 Based on the data of slopes (S) and drainage areas (A) of 118 gully-head catchments (table 1) we have developed an empirical power regression which can be used as geomorphic intrinsic threshold to gully 34 35 initiation on this lateritic terrain. The upstream slopes above gully heads are negatively correlated (r = -(0.55) with upstream drainage areas which are used as surrogate for the volume of runoff yield in the study 36 area. A significant line is fitted through the lower-most scatter points for the study sites which are incised to 37 form gully heads. This empirical straight line (S = 17.419 A  $^{-0.2517}$ , with R<sup>2</sup> of 0.52) represents an 38 approximation to critical slope - area threshold relationship for gully incision (figure 3). Any site (may be 39 40 un-trenched or trenched by gullies) lying above this critical line is much prone to gully erosion on this 41 terrain of laterites. It is derived that mean critical threshold slope for the initiation of gullies is 2.34°. The

high value of a (i.e. 17.419) signifies the initiation of gullies by high volume of overland flow and small 1 2 landslides in the study sites (Morgan and Mngomezulu, 2003). Most importantly the constant b is variously interpreted as relative area exponent or relative shear stress indicator (Bengin and Schumm, 1984; Morgan 3 4 and Mngomezulu, 2003). The negative value of b (i.e. -0.2517) and in general consideration b>0.2 is 5 considered to identify the dominancy of overland flow erosion over sub-surface processes in the study area 6 (Vandaele et al., 1996; Vandekerckhove et al., 1998; Morgan and Mngomezulu, 2003; Samni et al., 2009; 7 Dong et al., 2013). The slope – area relationship is recognized here as geomorphic threshold – intrinsic to the system to initiate abrupt changes as the primary condition of gully formation in this lateritic landscape. 8 The development of numerous gullies on laterites reflects geomorphic instability in the landform itself when 9 the critical hydro-geomorphic situation crossed the threshold limit, i.e.  $SA^{0.2715} > T$  (T is the threshold value, 10 i.e. 17.42 for this study site). It is estimated that critical drainage area for slope  $2.34^{\circ}$  is about 2908 m<sup>2</sup> to 11 12 initiate gully. Here we have compared our result of S - A threshold relation with the results of various studies conducted in a range of different environments (figure 4). It is found that our S - A critical line of 13 14 threshold is placed below the other lines, signifying a minimum geomorphic threshold to gully incision in 15 this tropical sub-humid monsoon climate and other geographical conditions.

16

20

17 To judge the slope – area relation (i.e. statistically fit or not) we have performed two statistical 18 techniques, viz., (1) Student's *t* test of correlation coefficient (*r*) and (2) significance test of standard error of 19 b ( $S_E$ ) (Sarkar, 2013).

Student's 
$$t = r\sqrt{(N-2)} / \sqrt{(1-r^2)}$$

21 Where *r* is Pearson product moment correlation coefficient, N is total number of sample and N - 2 is the 22 degree of freedom.

(Eq.3)

23 
$$S_E = b \sqrt{(1 - r^2)} / N$$
 (Eq.4)

24 Where the confidence limit of calculated  $S_E$  of b is  $(b + -1.96 S_E)$ .

The null hypothesis ( $H_Q$ ) is that there is no significant correlation between the two variables. For 116 25 degree of freedom (N - 2) the tabulated t value is 3.29 in 0.001 significance level (two-tailed) but our 26 27 calculated t value (7.09) much greater that tabulated t. Thus  $H_0$  is rejected and alternative hypothesis is accepted, which favours a significant inter-relation between S and A in the geomorphic system of gully 28 erosion. The calculated confidence limit of calculated  $S_E$  of b (0.271 to 0.232) does not include zero (i.e. 29 zero gradient). It signifies that the power regression(S = 17.419 A  $^{-0.2517}$ ) is certainly significant at five 30 percent level. Therefore, this slope - area threshold equation of channel initiation is valid statistically and 31 32 can be applied in the other erosion prone lateritic areas of *Rarh* Bengal (figure 5).

### 33 3.2 Model Validation and Application

We have tried to establish the S – A non-linear relationship (i.e. influence of intrinsic threshold) as a model to analyze the initiation criteria of gullies. The performance of this model is validated by the value of efficiency coefficient (EC) which was developed by Nash and Sutcliffe (1970) and this equation is applied successfully by Morgan and Duzant (2008) and Cao et al. (2013) in soil erosion research.

38 
$$EC = 1 - \Sigma (Q_{obs} - Q_{pred})^2 / (Q_{obs} - Q'_{obs})^2$$
 (Eq.5)

In the above equation  $Q_{obs}$  is measured value,  $Q_{pred}$  is calculated value and is  $Q'_{obs}$  mean of measured value.

Through inserting the values of drainage area ( $Q_{obs}$ ) in the equation of S = 17.419 A <sup>-0.2517</sup> the 1 predicated slope values (Q<sub>pred</sub>) of each gully is calculated. The mean slope of sample gullies (Q'<sub>obs</sub>) is 4.6°. 2 EC is estimated in the case of slope prediction and its value is greater than 0.63 (greater than 0.5) which is 3 4 generally interpreted to denote that this model performs satisfactorily (Morgan and Duzant, 2008). 5 Therefore, this model is validated in the study area. Then for experiment the S - A model is applied in the 82 gully heads of Masra – Jatla area (24°06'37'' to 24°08'15'' N, 87°39'38'' to 87°41'14'' E) and Bolpur – 6 Santiniketan area (23°40'47'' to 23°41'46'' N, 87°39'47'' to 87°40'36'' E) of Birbhum district. In this 7 badlands of laterites, we have derived two distinct threshold equations of  $S = 14.368 \text{ A}^{-0.236}$  (R<sup>2</sup> of 0.44) for 8 Masra – Jatla area and S = 112.48  $A^{-0.473}$  (R<sup>2</sup> of 0.85) for Bolpur – Santiniketan area respectively. In both 9 cases the dominancy of overland flow erosion is identified from significant b value (i.e. > 0.2). In these two 10 regions we have found that the value of EC varies from 0.54 to 0.77, depicting a good performance of S - A11 12 model.

### 13 **3.3 Dominancy of Overland Flow and M – D Envelope**

14 The trend line of A - S empirical relationship and regression slope (b value)can determine relative importance of overland flow erosion, subsurface flow erosion, diffusive erosion and mass movement or 15 16 landsliding erosion (figure 6). Here on the basis of slope (X axis) and drainage area (Y axis) we have 17 classified the gully heads to determine erosion dominancy which is clearly depicted through a threshold line, i.e. called Montgomery – Dietrich (M – D) Envelope. The estimated M – D envelope distinguishes mass 18 movement dominated gullies from hydraulic erosion dominated gullies. In this study area 52.51 percent of 19 gullies are affected by overland flow erosion (S  $- 1.2^{\circ}$  to 5.2° and A - 2129.1 to 10513.9 m<sup>2</sup>) while 27.96 20 percent belongs to landslide erosion (S - 5.2° to 9.5° and A - 457.1 to 5702.5 m<sup>2</sup>).Only 15.25 percent of 21 gullies (S - 2.2 to  $4.6^{\circ}$  and A - 685.5 to 3843.7 m<sup>2</sup>) are affected by tunnel erosion or seepage erosion (table 22 2). In the study sites the gullies are established by the deepening of rills and slumping of side slopes through 23 24 the shearing effect of concentrated overland flow, increase in pore-water pressure and decreases in soil strength along seepage lines close to the streams (Lal, 1992). Gully development in the vicinity of 25 concentrated flow is facilitated in the lateritic soils with predominantly coarse-textured A horizon (i.e. 26 27 secondary duricrust of loose ferruginous nodules) abruptly overlying a compact, less permeable mottle clay 28 or kaolinite pallid zone (B horizon). Therefore, based on the comparison with M - D envelope we can take preventive measures to check active processes in the gully sites. Also we can predict the un-trenched slope 29 30 facets which have chances to initiate gully heads on the laterites of *Rarh* Bengal. As this lateritic landscape 31 is affected by overland flow erosion, we can say that above the M - D envelope the excess runoff and 32 critical shear stress are progressively increased whereas below that line, the effect of rainfall intensity and 33 infiltration capacity is increased (Montgomery and Dietrich, 1994).

#### 34 **3.4 Relative Shear Stress**

Through inserting appropriate values of S and A for each sample site in the slope-area threshold 35 relation ( $\tau = S A^{0.2517} / 17.419$ , neglecting negative sign of b), we have estimated relative shear stress as 36 37 gradational threshold which is a geomorphic indicator of energy state expression of the gully system. The 38 result suggests a positive significant correlation (r = 0.92) between slope steepness (S) and relative shear 39 stress ( $\tau$ ) which is a ratio between shear stress applied by the flow and average shear stress of gully channel. In these experimental sites with increasing value of S, the magnitude of  $\tau$  is steadily increased with a linear 40 relation of  $\tau = 0.32675 + 0.4352$  (R<sup>2</sup> of 0.84). It signifies that to develop gully head the increasing slope 41 provides more kinetic energy to flow which generate more shear stress on the lateritic surface (figure 7). 42 This relation has importance to manage key factors of gully erosion. If we reduce the slope steepness and 43

drainage area through proper management (e.g. break in slope through terraces or ditches) above gully head, 1

we can trim down the shear stress of flow, flow accumulation and overland flow erosion. 2

#### 3.5 Estimation of Overland Flow through RMMF model and Critical Slope 3

5 6

4 The overland flow is acted as geomorphic extrinsic threshold to gully erosion, depending on effective rainfall, slope and other soil parameters. From the above analysis it is clear that overland flow (i.e. surface runoff) is foremost erosive agent above the gully head (i.e. catchment of gully head). Excess rain 7 water (i.e. water generated after satisfying the canopy interception and infiltration demand of the soil) contributes to surface runoff on the slope above gully head. We have applied the following equations of 8 9 modified RMMF of Morgan - Duzant version (Morgan and Duzant, 2008) to calculate annual overland flow on the sample slope angles and lengths of lateritic terrain. The details of used parameters are summarized in 10 11 table 3.

(Eq.7)

 $Rf = R (1 - PI) 1/\cos S$ 12 (Eq.6)

 $R_{c} = 1000 \text{ MS BD EHD } (E_{t}/E_{o})^{0.5}$ 

13

14

15 
$$Q = Rf \exp(-R_c/R_o) (L/10)^{0.1}$$
 (Eq.9)

16 Where, Rf is effective rainfall, R is mean annual rainfall, R<sub>o</sub> is ratio of mean rainfall to rainy days, PI is permanent interception by vegetation cover on slope, S is slope,  $R_c$  is soil moisture storage capacity, MS is 17 18 soil moisture content at field capacity, BD is bulk density of top soils, EHD is effective hydrological depth, is ratio of actual to potential evapotranspiration and L is slope length. 19

 $R_o = R / Rn (eq.8)$ 

It is found that 118 catchments of gully heads yield annual overland flow of 560.7 to 693.4 mm on 20 21 the laterites which have least growth of vegetation cover and ample portion of bare crusted ferruginous soils. This amount of overland flow is depended on the effective rainfall (Rf), slope and other soil hydrological 22 23 parameters of the region. The average overland flow of 619.5 mm is found to be sufficient to initiate gully on the critical slope angle and length. 24

It is derived from Du Boy's equation that the exerted eroding force of overland flow (mean depth of 25 flow is 0.0025 m) ranges from 0.58 to 5.32 N m<sup>-2</sup> above the gully heads. Here the slope – length ratio (S L) 26 is found to be important geomorphic variable of fluvial erosion to denote relative dominancy of high slope 27 28 with low length (i.e. high S L value) in the gully erosion. There is significant negative correlation of -0.69429 between S L and annual overland flow. The gully heads, with depth of 2.11 to 3.72 m, has high value of S L 30 (0.21 to 0.45). It reflects that those deep gullies on the laterites are formed due to steep angle of slope with relatively short slope length which provides high kinetic energy to mean overland flow of 560 mm. 31 Basically high S L with greater catchment is the most vulnerable site of gully erosion. Here overland flow is 32 empirically related with S L, developing a trend line of Q = 539.63 S L  $^{-0.0498}$  (R<sup>2</sup> of 0.65) (figure 8). Slope 33 length is found to be related with eroding force of overland flow, forming a critical trend line of F = -34  $0.41\log L + 2.464$  with R<sup>2</sup> of 0.49 (figure 9). 35

Twenty-eight un-trenched slope facets are plotted on that scatter diagram (figure 9) and same logic is 36 37 put here to drawn that threshold line, taking into consideration of lower most points. It is found that these slopes have critical situation regarding gully erosion prospect, because those points are located high above 38 the trend line. So these slopes on laterites are needed special attention and prevention to avoid initial rill and 39 40 gully formation. These vulnerable slopes vary in length from 72.2 to 221.6 m and in angle from 5.1° to

1 13.6°. The only safety factor of these sites is that the lateritic terrain is covered widely by bushes, grasses,

2 few tropical deciduous trees (mainly *Sal*) and *Acacia* plantation.

# 3 3.6 Estimation of Overland Flow Erosion Rate through RMMF model

RMMF model has the advantage to estimate annual rate of soil particle detachment by overland flow on the slope facet and this model is validated in the hill slopes of tropical region (Morgan, 2001). The net flow erosion is derived from the minimum value between annual rate of soil particle detachment by overland flow (H) and annual transporting capacity of overland flow (G) (if H < G then net annual erosion is H and vice versa).

9  $H = Z Q^{1.5} \sin S (1 - GC) 10^{-3}$  (Eq.10)

10 
$$Z = 1 / COH$$
 (Eq.11

11  $G = C Q \sin S 10^{-3}$  (Eq.12)

Where Z is soil erodibility constant, GC is ground cover, COH is soil cohesion and C is crop cover factor(table 3)

This analysis reveals that G is very high on this terrain, ranging from 8.8 to 72.3 kg m<sup>-2</sup> yr<sup>-1</sup> (table 1) 14 but present H ranges from only 2.33 to 19.9 kg m<sup>-2</sup> yr<sup>-1</sup>, i.e. annual rate of flow erosion in the sample 15 catchments. Here soil erosion is exceeded to general permissible limit (11.20 t ha<sup>-1</sup> yr<sup>-1</sup>). It is derived that 16 with increasing value of F the value of H is steadily increased in the slopes. This positive linear relation is 17 depicted through H =  $3.8266 \text{ F} + 0.6242 \text{ (R}^2 \text{ of } 0.9752)$ , having significant correlation coefficient of 0.98 18 19 (tested by Student's t) (figure 10 and table 4). Since the confidence limit of calculated  $S_E$  of b (0.645 to 0.602) does not enclose zero, then this relation is statistically valid for the lateritic region. The catchments 20 with high values of F are annually yield high amount of sediment (> 8 kg  $m^{-2} yr^{-1}$ ) due to overland flow 21 22 erosion. This analysis has vital aspect of erosion aggressiveness which is reflected in high potentiality of transport capacity of overland flow in the erosion prone gully catchments. 23

# 24 **4.** Conclusion

25 This research work highlights a crucial issue of gully erosion regarding intimate relation between critical valley slope and drainage basin area which can be used as predictive model of geomorphic instability 26 27 and it gives critical S- A value to locate those areas of instability within the valleys where gullies will form. 28 Here the recognition of significant geomorphic intrinsic and extrinsic thresholds (viz., slope, drainage area 29 and overland flow) is aided as practical and statistically validated approach to study gully erosion processes, 30 as a cause and effect analysis. With the influence of extrinsic threshold (Q) the instability of gully erosion system is finally triggered by the intrinsic threshold (A and S) which already exist within the system. We 31 have worked on existed permanent gullies which have high rate of head-cut migration (average 0.3 m yr<sup>-1</sup>) 32 and upslope flow erosion of up to 20 kg m<sup>-2</sup> yr<sup>-1</sup>. This phenomenon instigates rapid response from the 33 system as increasing severity of gully erosion, developing gully expansion through numerous initiations of 34 gully heads. Our quantitative study identifies the spatial dominance of overland flow over seepage and 35 landslide erosion to initiate gullies on the laterites. At last from the perspective of erosion management it can 36 be said that the established slope – area threshold relation (S = 17.419 A  $^{-0.2517}$ , S = 14.368 A  $^{-0.236}$  and S = 37 112.48 A<sup>-0.473</sup>) and other employed empirical models should be applied in the ravines and badlands of India 38 and in other lateritic region of Rarh Bengal to locate the vulnerable sites of gully erosion and to identify the 39 40 dominant processes of intensive erosion which should be checked fundamentally.

# 1 Acknowledgements

2 The research was funded by the University Grants Commission, New Delhi (Major Project No.: UGC-MRP-MAJOR-GEOG-2013-37968). We are very much thankful to Suvendu Roy (JRF, Dept. of 3 Geography, University of Kalvani), Subhankar Bera (JRF, Dept. of Geography, University of Kalvani) and 4 5 Subhamay Ghosh (M.Phil. Scholar, Centre for the Study of Regional Development, Jawaharlal Nehru University) for their rigorous all-round supports in the field study. We are sincerely grateful to all potential 6 7 reviewers for providing critical comments and suggestions on this research work. We are indebted to Prof. 8 Artemi Cerda (Professor, Dept. of Geography, University of Valencia) for giving us a great platform of 9 publication. We are very much grateful to Prof. R.P.C. Morgan (Emeritus Professor, Cranfield University), for providing invaluable articles and suggestions in this research work. 10

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
