# Peer review of "(untitled)"

_SOIL, 2016_

## Referee Comment (RC1) · Anonymous Referee #1 · 14 Sep 2016

Although the manuscript exhibit a high level of scholarship and original research that contributes to the field (especially the measured data needed in models), authors will need to undertake substantial grammatical revision and proof-reading. The authors in consultation with a qualified editor will need to undertake necessary grammatical revision.

Secondly, the manuscript provides context by stating that threshold conditions of gully initiation has not been performed in India. However, besides having a different location, the manuscript needs to provide more context with regards to other international studies, by stating (specifically) what about this study (methodology and/or results) are different or similar or applicable to other studies in other countries.

[Figure]

Some parts of the Results and Discussion, especially the sub-headings 3.5 and 3.6 should be rephrased to read as results and discussion (not read as methodology as is the case currently).

The soil erosion work cited by the authors are impressive, which include both new and older classic references. The authors correctly indicate that there are a wide range of threshold conditions or values (thresholds of hydraulic, rainfall, topography, lithology and land use – land cover control etc.) which are responsible for the initiation of gullies in different environments. Three other studies that did pioneering work in this regard includes:

Kirkby MJ, Bull LJ, Poesen J, Nachtergaele J, Vandekerckhove L. 2003. Observed and modelled distributions of channel and gully heads – with examples from SE Spain and Belgium. Catena 50 (2-4): 415-434. Desmet PJJ, Poesen J, Govers G, Vandaele K. 1999. Importance of slope gradient and contributing area for optimal prediction of the initiation and trajectory of ephemeral gullies. Catena 37: 377–392. Kheir RB, Wilson J, Deng Y. 2007. Use of terrain variables for mapping gully erosion susceptibility in Lebanon. Earth Surface Processes & Landforms 32: 1770–1782.

With so many citations the manuscript is probably exceeding the prescribed length of the journal. The first 3 paragraphs in Introduction (explains erosion as environmental problem) can be shortened.

---

## Referee Comment (RC2) · Anonymous Referee #2 · 18 Sep 2016

Despite my overall positive assessment of the article, I would like to suggest several adjustments which might contribute to the quality of the article.

1. The Introduction is very long. It contains information about the intensity of soil erosion processes around the world (South Asia, Europe, USA) as well as causes, intensity and results of soil erosion in relief transformation and agriculture in India. This section presents mainly quantitative data collected from numerous publications (ca 240) cited in the text. The title of the paper is "Geomorphic Threshold Estimation for Gully Erosion in the Lateritic Soil of Birbhum, West Bengal, India" but the discussion of threshold values covers an insignificant part of the text (p. 4, lines: 24-39; p. 5, lines: 9-18.) In my opinion Introduction should be shortened but the fragment about threshold

values should be more detailed. I would like the authors to explain how they understand the terminology they use in the text: geomorphic threshold, threshold value, threshold condition, critical threshold conditions , geomorphic threshold estimation, threshold relation, threshold phenomenon, critical hydro-geomorphic situation, threshold limit, threshold line in view of study of gully development of Birbhum, West Bengal, India.

2. The authors admit that in determining thresholds of gully development in India they make use of methods devised by other researchers and implemented earlier in other regions of the world. I would suggest supplementing the text with the discussion of the results obtained by other authors. Are they much different and if so why. In figure captions one should quote not only the area to which a given line applies but provide the name of the author and year of publication (fig.4).

3. In section 3.5 there is no information about the length of the measurement period of precipitation on the basis of which the overland flow was calculated.

4. Suggestions about the figures: In my opinion the information included in figures 4, 6, and 9 was not fully interpreted especially with reference to the results obtained by other authors. Figure 8: The description of axis should be done with the fonts used for other figures.

5. Suggestions about the "Reference cited": The items in the reference list are not all compiled in the same style (p. 12, lines:13-15,17, 19, 21,24-25, 26, 29, 30, 34, 35; p.13, lines: 2-3, 7, 10, 13-15, 16, 19, 21, 22, 28, 29,32, 35, 37-38; p.14, lines: 2, 3-4, 6, 9, 12, 14, 15, 17, 20, 23, 24-25, 27, 30, 34, 35, 37; p. 15, lines: 3-4, 6, 8, 12-13, 15, 18, 21-22, 23-24, 25, 31, 33, 36; p. 16, lines:3-4, 6, 10, 13, 16-18, 20, 22, 25, 32, 33, 36; p.17, lines: 1-2, 4, 6-7, 9, 11, 13, 14, 16-17, 19-20, 23, 26, 28, 30, 35, 37; p. 19, lines: 1, 4, 6, 9, 10-11,13, 16, 22-23, 26, 28, 29, 31, 33, 35; p. 19, lines: 1, 4, 7. It refers mainly to different styles used, spaces and dots.

I hope that my remarks will help the Authors improve their article. I do appreciate their diligence in collecting the data, which were used to establish threshold conditions

in permanent gully development in Birbhum, West Bengal, which are an important element in the study of gully erosion mechanism.

---

## Author Comment (AC1) · 4 Nov 2016

Journal - SOIL Article Title – Geomorphic Threshold Estimation for Gully Erosion in the Lateritic Soil of Birbhum, West Bengal, India Doi – 10.5194/soil-2016-48, 2016 Authors – Sandipan Ghosh and Sanat Kumar Guchhait Responses of Comments of Referee #1 Comment 1: Although the manuscript exhibit a high level of scholarship and original research that contributes to the field (especially the measured data needed in models), authors will need to undertake substantial grammatical revision and proof-reading. The authors in consultation with a qualified editor will need to undertake necessary grammatical revision.

Reply 1: We are very much pleased that our research work is getting importance in the

publication forum of EGU – SOIL. It is nice to observe that the referee found interest in this manuscript and he appreciated our scholarly work. The referee recommended us to revise the manuscript including grammatical checks. We have now tried to revise the manuscript again to undertake necessary corrections. Comment 2: Secondly, the manuscript provides context by stating that threshold conditions of gully initiation has not been performed in India. However, besides having a different location, the manuscript needs to provide more context with regards to other international studies, by stating (specifically) what about this study (methodology and/or results) are different or similar or applicable to other studies in other countries.

Reply 2: If we search in the website, surely we can't found any kind research regrading to measure the threshold conditions of gully initiation in Indian terrain. Though, the soil loss and land degradation due to gully erosion is an emerging environmental issue in this agricultural dependent country. Now we have included a section of Introduction about the methodology and results of other erosion researches in comparison to India. Though, the methodology section has a major part of the manuscript. Comment 3: Some parts of the Results and Discussion, especially the sub-headings 3.5 and 3.6 should be rephrased to read as results and discussion (not read as methodology as is the case currently).

Reply 3: The sections of 3.5 and 3.6 are under Results and Discussion. 3.5 section is concentrated on the estimation of overland flow on the slopes and 3.6 section discusses the result of RMMF model derived erosion rate in the slopes. In this section it is necessary and logical to include the model derived equations which are applied to calculate the overland flow and annual erosion rate. For that reason we have included a small section of quantitative expressions. Then the other parts include only results and trend.

Comment 4: The soil erosion work cited by the authors are impressive, which include both new and older classic references. The authors correctly indicate that there are a wide range of threshold conditions or values (thresholds of hydraulic, rainfall, topography, lithology and land use – land cover control etc.) which are responsible for the initiation of gullies in different environments. Three other studies that did pioneering work in this regard includes:

Kirkby MJ, Bull LJ, Poesen J, Nachtergaele J, Vandekerckhove L. 2003. Observed and modelled distributions of channel and gully heads – with examples from SE Spain and Belgium. Catena 50 (2-4): 415-434. Desmet PJJ, Poesen J, Govers G, Vandaele K. 1999. Importance of slope gradient and contributing area for optimal prediction of the initiation and trajectory of ephemeral gullies. Catena 37: 377–392. Kheir RB, Wilson J, Deng Y. 2007. Use of terrain variables for mapping gully erosion susceptibility in Lebanon. Earth Surface Processes & Landforms 32: 1770–1782.

Reply 4: The referee admired that we reviewed and cited good referenced works of past and present. The referee suggested to include three another important referenced works in the manuscript. Therefore, we have included these. Comment 5: With so many citations the manuscript is probably exceeding the prescribed length of the journal. The first 3 paragraphs in Introduction (explains erosion as environmental problem) can be shortened.

Reply 5: We have shortened the first 3 paragraphs in the section of Introduction. Alongside we have tried to delete few less significant references.

Responses of Comments of Referee #2 Comment 1: The Introduction is very long. It contains information about the intensity of soil erosion processes around the world (South Asia, Europe, USA) as well as causes, intensity and results of soil erosion in relief transformation and agriculture in India. This section presents mainly quantitative data collected from numerous publications (ca 240) cited in the text. The title of the paper is "Geomorphic Threshold Estimation for Gully Erosion in the Lateritic Soil of Birbhum, West Bengal, India" but the discussion of threshold values covers an insignificant part of the text (p. 4, lines: 24-39; p. 5, lines: 9-18.) In my opinion Introduction should be shortened but the fragment about threshold values should be more detailed.

I would like the authors to explain how they understand the terminology they use in the text: geomorphic threshold, threshold value, threshold condition, critical threshold conditions , geomorphic threshold estimation, threshold relation, threshold phenomenon, critical hydro-geomorphic situation, threshold limit, threshold line in view of study of gully development of Birbhum, West Bengal, India.

Reply 1: As suggested by the referee we have tried to shorten the section of Introduction. It is suggested to give more statements about threshold in introduction. So, we have now tried to give more focus on the different dimensions of geomorphic threshold. We don't prescribed terminology related to only threshold which is very dynamic concept in respect to physical geography. In this work we have only geomorphic dimension where threshold is placed in the dynamic – process geomorphology. So, the terminology of geomorphic threshold, threshold value, threshold condition etc. is not depicted to present diverse explanation or situation. We have used the term 'threshold' to define a critical condition scale (or value) from where a new equilibrium has occurred in the geomorphic system (i.e. erosion system). So to explain the facts of different results we have used the terminology of threshold as per requirement. For example, based of slope and drainage area the threshold line of erosion system denotes here the perfect non-linear regression line (with high coefficient of determination) above which the initiation of gully starts in the laterite terrain. Comment 2: The authors admit that in determining thresholds of gully development in India they make use of methods devised by other researchers and implemented earlier in other regions of the world. I would suggest supplementing the text with the discussion of the results obtained by other authors. Are they much different and if so why. In figure captions one should quote not only the area to which a given line applies but provide the name of the author and year of publication (fig.4).

Reply 2: We have tried to include the results of other research works in comparison to our work. We have now written the author citations in figure 4. Comment 3: In section 3.5 there is no information about the length of the measurement period of precipitation

on the basis of which the overland flow was calculated.

Reply 3: We have used rainfall data of 2015 (i.e. mean of 1990 to 2015) which was provided by Irrigation and Waterways Department, Government of West Bengal, India. Based on this data we have calculated the overland flow. Comment 4: Suggestions about the figures: In my opinion the information included in figures 4, 6, and 9 was not fully interpreted especially with reference to the results obtained by other authors. Figure 8: The description of axis should be done with the fonts used for other figures.

Reply 4: We have now revised the interpretation of figure 4, 6 and 9 in the text. The description of axis is included the figure 8. Comment 5: Suggestions about the "Reference cited": The items in the reference list are not all compiled in the same style (p. 12, lines:13-15,17, 19, 21,24-25, 26, 29, 30, 34, 35; p.13, lines: 2-3, 7, 10, 13-15, 16, 19, 21, 22, 28, 29,32, 35, 37-38; p.14, lines: 2, 3-4, 6, 9, 12, 14, 15, 17, 20, 23, 24-25, 27, 30, 34, 35, 37; p. 15, lines: 3-4, 6, 8, 12-13, 15, 18, 21-22, 23-24, 25, 31, 33, 36; p. 16, lines:3-4, 6, 10, 13, 16-18, 20, 22, 25, 32, 33, 36; p.17, lines: 1-2, 4, 6-7, 9, 11, 13, 14, 16-17, 19-20, 23, 26, 28, 30, 35, 37; p. 19, lines: 1, 4, 6, 9, 10-11,13, 16, 22-23, 26, 28, 29, 31, 33, 35; p. 19, lines: 1, 4, 7. It refers mainly to different styles used, spaces and dots.

Reply 5: The problematic reference citations are now corrected in the text as same style.